# The Physiological and Molecular Mechanisms of Silicon Action in Salt Stress Amelioration

**DOI:** 10.3390/plants13040525

**Published:** 2024-02-15

**Authors:** Siarhei A. Dabravolski, Stanislav V. Isayenkov

**Affiliations:** 1Department of Biotechnology Engineering, Braude Academic College of Engineering, Snunit 51, Karmiel 2161002, Israel; sergedobrowolski@gmail.com; 2International Research Centre for Environmental Membrane Biology, Foshan University, Foshan 528000, China; 3Institute of Agricultural and Nutritional Sciences, Martin Luther University Halle-Wittenberg, Betty-Heimann-Strasse 3, 06120 Halle, Germany; 4Department of Plant Food Products and Biofortification, Institute of Food Biotechnology and Genomics, The National Academy of Sciences of Ukraine, Baidi-Vyshneveckogo Str. 2a, 04123 Kyiv, Ukraine

**Keywords:** salinity stress, silicon transport, stress amelioration, cell wall, membrane transport, stress regulation

## Abstract

Salinity is one of the most common abiotic stress factors affecting different biochemical and physiological processes in plants, inhibiting plant growth, and greatly reducing productivity. During the last decade, silicon (Si) supplementation was intensively studied and now is proposed as one of the most convincing methods to improve plant tolerance to salt stress. In this review, we discuss recent papers investigating the role of Si in modulating molecular, biochemical, and physiological processes that are negatively affected by high salinity. Although multiple reports have demonstrated the beneficial effects of Si application in mitigating salt stress, the exact molecular mechanism underlying these effects is not yet well understood. In this review, we focus on the localisation of Si transporters and the mechanism of Si uptake, accumulation, and deposition to understand the role of Si in various relevant physiological processes. Further, we discuss the role of Si supplementation in antioxidant response, maintenance of photosynthesis efficiency, and production of osmoprotectants. Additionally, we highlight crosstalk of Si with other ions, lignin, and phytohormones. Finally, we suggest some directions for future work, which could improve our understanding of the role of Si in plants under salt stress.

## 1. Introduction

Salinity stands as a formidable abiotic stressor significantly impacting global agricultural productivity, with detrimental effects observed in at least 20% of crop cultivation worldwide [1]. When subjected to elevated saline conditions, plants experience a drastic reduction in water absorption rates, disrupting both inter- and intracellular water levels and culminating in inhibited cell expansion and decreased stomatal activity. Prolonged exposure to salinity induces substantial ionic and oxidative stresses, primarily attributed to increased NaCl influx. The resulting ionic and osmotic imbalances under salt stress compromised overall plant growth and development [2]. Salt accumulation further decreases essential photosynthetic pigments like chlorophyll and carotenoids, disrupting ribulose-1,5-bisphosphate functionality and degrading the photosynthetic apparatus. This cascade effect results in an overproduction of reactive oxygen species (ROS) that surpasses the plant’s natural scavenging capacity. Excessive ROS levels impede transpiration and nutrient uptake and inflict damage on vital macromolecules, including nucleic acids, proteins, and lipids, ultimately compromising membrane integrity and essential metabolic processes [3].

Moreover, the effects of NaCl on protein synthesis, enzyme activities, and photosynthesis contribute to chlorosis, necrosis, and premature senescence of older leaves [4]. To overcome the negative effects of salt stress, plants apply various mechanisms such as ion exclusion, osmotic tolerance, redox homeostasis, and efficient photosynthesis. Researchers across various plant biology fields are actively engaged in initiatives aimed at unravelling the intricate responses to salinity stress, with the goal of developing stress-tolerant crop varieties and advancing sustainable agricultural practices in saline environments [5].

Silicon (Si) is the second most abundant element in the Earth’s crust. Traditionally viewed as a beneficial but nonessential element for plant growth, Si has recently earned recognition as a “quasi-essential” element [6]. The supplementation of Si has demonstrated various benefits for plants, including increased yield [7], enhanced disease resistance, and mitigation of abiotic stresses [8,9]. Si supplementation has proven effective against diverse stresses such as salinity [10,11], drought [12], high temperatures [13], hyperhydricity [14], and various pathogens [15].

The mechanism through which Si confers stress resistance in plants involves several physiological improvements, including enhanced growth and biomass, effective nutrient management, maintenance of structural rigidity, improved photosynthetic efficiency, resistance to lodging, ion homeostasis balance, activation of antioxidant systems, induction of stress-related secondary metabolites, and regulation of genes associated with various biochemical and physiological processes [16]. These merits of Si have been harnessed in diverse crops to increase yield and enhance tolerance to different stresses. While numerous studies have demonstrated the positive impacts of Si supplementation, the exact molecular mechanisms underlying stress alleviation are still under investigation. In this review, we aim to provide a summary of the recently revealed mechanisms elucidating how silicon alleviates salinity stress.

## 2. Mechanisms of Si Uptake, Transport, and Accumulation

Si predominantly exists in the soil in the forms of silica and silicates, while plants can primarily uptake and utilise orthosilicic acid Si[OH]_4_, thus keeping most of Si inaccessible to plants [17]. The process of slow gradual release of Si from soil minerals may be exacerbated by the improper application of fertilisers and poor agrotechnology, leading to decreased availability of Si for plant absorption and subsequently causing deficiency of orthosilicic acid in soils [18]. Plants uptaking high Si levels have been classified as active (for example, such monocots as rice, wheat, and barley) (more than 1% of dry weight in aboveground parts), intermediate accumulators as passive (such as oat) (accumulation from 0.5% to 1%), and plants with lower uptake as rejective (tomatoes) (accumulated below 0.5%) [19,20].

Several studies have revealed the intricate mechanisms underlying Si uptake and transport in plants. An original Si transporter was identified in rice and affiliated with one of the aquaporin (AQP) subfamily—the nodulin 26-like intrinsic proteins (NIPs) [21]—and the family of the membrane intrinsic proteins (MIPs), which manifested highly conserved features, such as the NPA (asparagine–proline–alanine) domain and ar/R (aromatic/arginine) selectivity filter [22]. Subsequently, Si transporters have been identified in other mono- and dicot species (such as rice [23], maize, barley [24], wheat [25], pumpkin [26], grape [27], cucumber [28,29], tomato [30], and many others [31]).

Numerous studies have revealed the structural and functional details of plant Si transporters. Rice, as a species of significant agricultural importance and an active Si accumulator, was used as the first model species to define the structure of the Si transporter—low silicon rice 1 (Lsi1) [32,33]. The narrowest part of the channel mainly determines its substrate selectivity. This so-called four amino acid selectivity filter (SF) was used to further classify the transporters’ subfamilies into several subgroups [34]. Sequence analysis and mutagenesis experiments have demonstrated that Lsi1 belongs to the NIP-III subgroup with conserved GSGR amino acids of SF, which are crucial for silicic acid permeation and common in high Si-accumulating plants. On the other hand, low accumulators have a more diverse amino acid composition of the SF (such as AVAR, FAAR, WIGR, WVAR, and AIGR) [35]. Substrate transport in Lsi1 is passive and bidirectional, driven by concentration gradients. In addition to Si, Lsi1 is also involved in the transport of boric acid and arsenite [36]. The mystery of Lsi1’s effective Si transport, high selectivity, and accumulation was clarified by the solved crystal structure, where four SF amino acids (GSGR) and a fifth amino acid (Thr65), which binds two unique water molecules (Water3 and Water9), have been shown to play the crucial role [32,33]. Another feature was the presence of exactly 108 amino acids between two highly conserved NPA domains, which form the second constriction to the pore [37]. Interestingly, the Si transporter from tobacco (*Nicotiana sylvestris* Speg. & Comes) possesses both features specific for high Si accumulators, while it is known as a low Si accumulator. The *NsLsi1* expressed constitutively in planta, and the analysis of its amino acid sequence and comparison with rice Lsi1 suggested that NsLsi1^P125F^ may be responsible for a functional defect and, subsequently, low Si accumulation. Point mutation NsLsi1^P125F^ displayed a three times increase in Si influx when compared to NsLsi1^WT^ and coincided with a three times increase in plasma membrane localisation in planta [38]. 

Recently, the importance of both N- and C-terminal regions for polar OsLsi1 localisation was shown. Thus, the deletion of both N- and C-terminal parts leads to the loss of OsLsi polar localisation. Also, a cluster of positively charged residues at the C-terminal tail was crucial for the proper localisation because the deletion of the C-terminus inhibited OsLsi trafficking from the ER to the plasma membrane. Particularly, the Ile18 and Ile285 (at the N-terminal and C-terminal region, respectively) were crucial for the OsLsi1 polar localisation and efficient Si uptake [39]. Furthermore, the rice Si transporter OsLsi3 was characterised as responsible for Si loading into the xylem. *OsLsi3* was mainly expressed in the mature region of the root and localised to the pericycle without polarity (Figure 1). Interestingly, Si supply downregulated *OsLsi3* expression, while its knock-out reduced Si uptake and xylem sap concentration under low Si supply but not under high Si supply [40]. 

While many studies have characterised influx Si transporters (Lsi1), much less is known about efflux transporters (Lsi2). Thus, two Si efflux transporters genes (*EaLsi2-1* and *EaLsi2-2*) were identified in horsetail (*Equisetum arvense* L.), which is known as one of the highest Si accumulators among plants. These two genes showed low sequence similarity with their homologues in higher plants, were localised to the plasma membrane, and were fully functional when expressed in *Xenopus* oocytes. Interestingly, the expression level in shoots was much higher than in the roots and the availability of Si did not affect the expression level [41]. Similarly, grape (*Vitis vinifera* L.) aquaporin *VvNIP2;1* is highly expressed in roots and green berries and is localised at the plasma membrane. Experiments in *Xenopus* oocytes confirmed that VvNIP2;1 was responsible for Si and arsenite As^III^ anion uptake [27].

The function of Si transporters was recently characterised in tomato (*Solanum lycopersicum* L.). Interestingly, the tomato’s homologue of rice *Lsi1* was constitutively expressed in the roots, localised at the plasma membrane of both the root tip and basal region without polarity, and was fully functional when expressed in heterologous systems (rice *lsi1* plants and *Xenopus laevis* oocytes). SlLsi2-like protein, on the contrary, did not show any Si efflux activity in *Xenopus* oocytes, while Lsi2 transporter from cucumber (CsLsi2) was functional in tomato and significantly increased Si uptake and accumulation, thus enhancing tolerance to abiotic stresses. These results explained low Si accumulation in tomatoes because this species lacks the functional Si efflux transporter (Lsi2 homologue), although the Si influx transporter (Lsi1 homologue) is fully functional [30]. 

### 2.1. Si Accumulation

Microscopic analysis revealed that in *Cannabis sativa* L., Si accumulated in the distal cell walls of bast fibres and the basal cells of leaf trichomes, thus suggesting also a mechanical function in supporting the trichome shaft [42]. Interestingly, Si deposition varied significantly even among members of one family. Thus, in leaves of cucumber (*Cucumis sativus* L.), pumpkin (*Cucurbita maxima* Duchesne), and melon (*Cucumis melo* L.), the high accumulation of Si was detected in cells surrounding the bases of the trichome hair, while the hair itself deposited calcium. On the contrary, in sponge gourd (*Luffa cylindrica* M.Roem.) and bottle gourd (*Lagenaria siceraria* (Molina) Standl. var. *hispida*), high accumulation of Si was detected only in the hair. In watermelon (*Citrullus lanatus* (Thunb.) Matsum. & Nakai) leaves, Si was deposited in both the hair and cells surrounding the bases of the hair [43]. 

Further research on Si accumulation in epidermal silica cells of grass species sorghum (*Sorghum bicolor* (L.) Moench) demonstrated that Si deposition is an active, physiologically regulated process that requires viable cells and is independent of water evapotranspiration [44]. Specifically, the protein Siliplant1 (Slp1) was shown to play a crucial role in Si metabolisation in silica cells. Slp1 RNA was found in immature leaves and immature inflorescence, and the Slp1 protein was localised into the developing silica cells, packed inside vesicles and diffused throughout the cytoplasm. Upon fusing with the cell membrane, their content is released into the apoplastic space where it interacts with supersaturated silicic acid solution and causes silica precipitation. Subsequently, the growing siliceous cell wall constricted, thus reducing the cytoplasmic volume of silica cells and resulting in programmed cell death, which ended the silicification process [45]. 

Interestingly, in tree species date palm (*Phoenix dactylifera* L.), Si was accumulated in specific stegmata cells found in roots, shoot apex, leaf petioles, and blades, and their surfaces were composed of pure silica. These stegmata cells were abundant on the outer surface of the sclerenchyma bundles (fibres), thus also suggesting a mechanical function. However, in contrast to grasses, where Si deposition was mostly associated with cell walls, the silica phytoliths in palms appear to be formed intracellularly [46].

Fascinatingly, one of the current hypotheses suggested that plants utilise Si as a structural component to enhance their strength and rigidity and protect them from pathogens. Also, structural Si may serve as an alternative to lignin biosynthesis because it is more resource-demanding [47]. Some studies have supported this hypothesis: (1)Negative associations between Si and lignin content in several species [48];(2)Increased lignin biosynthesis in rice Si transporter mutants [49];(3)Absence of Si in hydroponic medium induced cell wall thickening in rice. The expression of lignin biosynthesis-related genes and secondary cell wall cellulose synthase genes was upregulated [50];(4)In the rice straw from many locations across South East Asia, the concentrations of Si were negatively related to the concentrations of carbon and lignin-derived phenols [51].

The composition and structure of lignins in the cell walls of sorghum seedlings were analysed in hydroponic conditions with or without Si supplementation. Limited Si supply increased the ratio of the thioacidolysis-derived syringyl/guaiacyl monomer and the content of thioglycolic acid lignin, which, at least partially, could be explained by the upregulated expression of many phenylpropanoid biosynthesis-related genes. Also, plants growing under limited Si supply (−Si) contained 31% more lignin than +Si plants; in particular, the content of G and S lignin monomers was increased. Furthermore, the cell walls of the −Si plants showed higher mechanical strength and calorific value compared to the +Si plants. Interestingly, the expression of genes responsible for callose metabolism (*glucan synthase-like* and *glucan 1,3-β-glucosidase*) was upregulated under −Si conditions, while the expression of genes responsible for the metabolism of other cell wall polysaccharide components was not affected [52].

Further research confirmed that the active uptake of silicic acid ([H_2x_SiO_x+2_]_n_) occurred at the root apex, where *Lsi1* and *Lsi2* are expressed, while silica aggregation occurred in nonlignified spots in the endodermal cell walls, where silicic acid accumulates and condensates at arabinoxylan–ferulic acid complexes, thus hampering further lignin deposition. Interestingly, the silica aggregation sites are established independently of the presence of silicon but are predetermined by a cell wall architecture and governed by endodermal development [53]. Furthermore, silica aggregates form in an orderly pattern of spots (deposition of modified lignin and silica) along the inner tangential cell walls of root endodermis cells. In −Si conditions, the chemically modified lignin crosslinked into the cell wall and lost its ability to nucleate silica. On the contrary, in +Si conditions, silica polymerised on the modified lignin, thus affecting the chemistry of lignin [54]. Indeed, experiments in vitro confirmed that silica aggregates were deposited onto freshly polymerised coniferyl alcohol (simulating G-lignin) but not onto monomers of coniferyl alcohol or ferulic acid [55]. Further study showed that the apoplastic levels of oxidative stress (in the form of H_2_O_2_) regulate the extent of endodermal silicification through the mediation of lignin-like formation—the active silicification zone (ASZ). Thus, oxidative stress increased deposition of the lignin-like ASZ via increased activity peroxidases and H_2_O_2_, thus enhancing silification. The background lignin, however, is deposited independently of oxidative status and does not affect silica aggregation [56]. 

The interaction between silica and lignin requires further investigation. Mainly, it is important to prove if the site-specific silicification has developed as a response to adverse environmental factors (such as malnutrition and biotic or abiotic stresses) or as a protective mechanism against potential Si toxicity mediated via competition between silica and lignin. The answer to this fundamental question would greatly advance our understanding of Si metabolism and its role in plant productivity.

### 2.2. Crosstalk between Si and Other Ions

Recent experiments on hemp (*Cannabis sativa* L.) demonstrated that salt stress stimulated Si uptake, which subsequently accumulated in leaves and ameliorated the symptoms of salinity. Interestingly, the expression of the hemp orthologous of rice *Lsi2* was increased in response to salt stress but not to Si supplementation. At the same time, Si supplementation did not significantly affect the expression of genes responsible for salt stress [57]. A recent study demonstrated that in addition to Si, iron (Fe) availability in the root growth environment is also crucial for *Lsi1* transcript accumulation [58]. On the other side, Si accumulation in the rice shoots was shown to suppress the Zn uptake (Figure 2). Si supplementation reduced Zn concentration in the roots and shoots of wild-type plants but not in the *lsi1* plants. Thus, Si accumulation in the shoots decreased ^67^Zn isotope uptake in *lsi1* plants but no root-to-shoot transport in the wild type. Indirectly, this effect was mediated through the downregulation of Zn transporter *OsZIP1* in the wild type, while its expression in *lsi1* was not altered [59]. 

Si supplementation (in the form of potassium silicate (K_2_SiO_3_)) of different poinsettia (*Euphorbia pulcherrima* Willd. ex Klotzsch) cultivars (a low Si-accumulator species) increased the content of Mg and decreased the content of B and Zn in the roots, and only the content of sulphur (S) was increased in the shoots [60]. At the same time, short-term Si supplementation decreased the expressions of Si transporters (*EpLsi1* and *EpLsi2*) in the roots and leaves. Long-term Si supplementation, on the contrary, increased *EpLsi1* expression in the leaves, bracts, and cyathia, while *EpLsi2* expression increased only in the roots and leaves, suggesting that sufficient Si supplementation can increase tissue Si content [61]. 

Priming the rice seeds with As^(III)^ + Si helped plants tolerate As stress for longer. In particular, the As + Si treatment increased the expression of *OsLsi1/2/6* genes in both shoots and roots and resulted in lower As accumulation in the presence of Si. Also, the expression of the main genes responsible for the absorption and assimilation of N (*nitrate reductase* (*NR*), *nitrite reductase* (*NiR*), *glutamine synthetase* (*GS*), *glutamate synthase* (*GOGAT*), *high-affinity nitrate transporter protein* (*NRT2*) and *high-affinity ammonium transporter protein* (*AMT1*), P (*phosphate transporter* (*PT*), *high-affinity phosphate transporter 1* (*PHT1*), *high-affinity phosphate transporter 2* (*PHT2*), and *acid phosphatases* (*APase*)) and K (*potassium channel protein* (*KAT1*) and *potassium transporter protein* (*HAK10*)) was significantly increased by every variant of treatment (As, Si, and As + Si) (Figure 2). The malondialdehyde (MDA) content was increased by all variants of treatments (As, Si, and As + Si), while the activity of superoxide dismutase (SOD), catalase (CAT), glutathione peroxidase (GPX) and glutathione S-transferase (GST) was stimulated only by As and As + Si but not by Si. These results suggested that priming seeds with Si is beneficial for As^(III)^-stressed rice plants [62].

## 3. Effect of Si on Plant Physiology and Biochemistry

Si application was beneficial for the growth and photosynthetic performance of cucumber plants under salt stress. In particular, the expression of 1469 genes involved in ion transport, hormone biosynthesis (jasmonic and salicylic acid, ethylene), signal transduction, biosynthetic and metabolic processes, responses to stress, and defence were altered by Si treatment. These results suggest Si may act as an important mediator in inducing salt tolerance in cucumber plants [63]. Similarly, the application of Si on in vitro-grown *Rosa hybrida* alleviated the negative effects of salt. The analysis of proteomic data demonstrated Si increased the abundance of 30 proteins and downregulated five proteins. The major protein groups affected by Si were functionally classified as photosynthesis, carbohydrate/energy metabolism, and transcription/translation [64]. 

Furthermore, metabolomic analysis of salt-stressed date palm (*Phoenix dactylifera* L.) seedlings showed Si affected the accumulation of 1101 metabolites in leaves and roots. The most important affected groups were antioxidants (allithiamine, boldine, cepharanthine, myristic acid, and pyridoxine), osmoregulators (such as mucic acid, glucaric acid, saturated myristic fatty acid, two methylated derivatives of myo-inositol, soluble sugar L-pinitol, and (-)-bornesitol), and detoxification intermediates (such as gamma-glutamyl conjugates, beta-cyano-L-alanine, and S-D-lactoylglutathione). Importantly, Si application enhanced the formation of the Casparian strip [65]. Moreover, metabolic investigation of the Si effect on salt-stressed tomato plants (*Solanum lycopersium* L.) demonstrated that the beneficial effects were mediated through modulation of the primary metabolic pathways involved in the biosynthesis of several amino acids, the fatty acid and tricarboxylic acid cycle, and the phenylpropanoid and flavonoid biosynthesis pathways [66]. 

Identification of the differentially expressed genes and accumulated metabolites could offer valuable insight into how Si can promote growth and enhance salt tolerance in plants. Further in this section, we discussed the effect of Si supplementation on various systems (metabolism of different ions, hormones, osmoprotectants, antioxidants, and photosynthesis) in salt-stressed plants. 

### 3.1. Uptake and Transport of Na^+^, K^+^, and Cl^−^

The role of Si in mitigating the detrimental effects of salt stress and augmenting salt tolerance in plants has been extensively investigated; however, the underlying mechanisms remain somewhat elusive. Si supplementation is mostly attributed to enhanced salt tolerance through reduced plant susceptibility to damage from osmotic stress and ionic toxicity (Figure 2). Several mechanisms of Si-mediated salt tolerance have been observed in various plants. Thus, the major Si-involving mechanism in conferring salt tolerance was recognised as diminishing Na^+^ uptake and maintaining an optimal Na^+^/K^+^ ratio [67]. Si application improves the uptake of K^+^ and Ca^2+^ over Na^+^ uptake, fostering enhanced growth and productivity in sunflower (*Helianthus annuus* L.) and sorghum (*Sorghum bicolor* L. ‘Moench’) [68]. Similarly, under salt stress conditions, Si application (both foliar and root) decreased Na^+^ uptake and increased nutritional efficiency, thus improving the total plant dry weight of both sorghum and sunflower plants [69]. 

Experiments on wheat showed that salt stress increased Na^+^ and Cl^−^ concentrations in root epidermal, cortical, and stellar cells while decreasing in the vascular bundle cells. K^+^ and Mg^2+^ profiles, on the contrary, were opposite to those of Na^+^ and Cl^−^ (Figure 2). These salt stress-mediated changes in ion profiles were correlated with decreased chlorophyll content, photosynthesis rate, and enhanced electrolyte leakage index, and they ultimately impaired plant growth. However, Si supplementation improved plant growth and performance under salt stress by reducing Na^+^ and Cl^−^ ion levels in root epidermal and cortical cells and, by increasing root uptake, the storage and xylem loading of K^+^ and Mg^2+^ ions [70]. 

Further insights from rice suggested that Si affects Na^+^ translocation via bypass flow, a heritable trait associated with leaf Na^+^ accumulation and salt sensitivity in some rice cultivars [71]. Si promoted plant growth under salt stress by reducing the bypass flow (root-to-shoot Na^+^ translocation) in wild types but not in *lsi1* and *lsi2* plants. Also, Si was accumulated at the root endodermis of wild-type, but not mutant, plants. Also, Si treatment downregulated the expression of *OsLsi1* and *OsLsi2* genes in the root, thus accelerating Si deposition at the root endodermis [72]. Thus, experiments with radiotracer ^24^Na^+^ showed Si lowered Na^+^ in shoots of both salt-sensitive and salt-tolerant rice cultivars, while the root and shoot K^+^ levels were not affected. However, Si lowered shoot Na^+^ in the salt-sensitive cultivar via a great reduction in bypass flow, while in the salt-tolerant cultivar, because the bypass flow was small and not affected by Si treatment, it occurred through a growth dilution of shoot Na^+^. Interestingly, Si supplementation did not affect Na^+^-stimulated plasma membrane depolarisation or unidirectional ^24^Na^+^ fluxes (influx and efflux) in both cultivars [73].

However, contradicting results were reported in other research investigating the effect of Si supplementation on ion homeostasis in rice. Therefore, Si supplementation improved salt stress-induced K^+^ deficiency by increasing K^+^ uptake index and shoot distribution rate and enhanced K accumulation content and concentration in xylem sap of wild-type, but not *lsi1* and *lsi2*, plants. Also, Si upregulated the expression of genes responsible for K^+^ uptake and xylem loading (*AKT1*, *HAK1*, and *SKOR*), thus increasing the K^+^ influx rate under salt stress only in wild-type plants (Figure 2) [74].

Furthermore, salt overly sensitive 1 (OsSOS1), a Na^+^/H^+^ antiporter involved in the maintenance of ion homeostasis and salinity response, was also found to correlate with Si nutrition (Figure 2). Thus, the expression of *Lsi1* and *Lsi2* was increased in *SOS1* overexpressing plants under salt stress compared to the wild type. Also, Si-supplemented transgenic plants under salt stress accumulated more K^+^ and Ca^2+^ and less Na^+^ and Cl^−^ compared to wild-type plants [75]. These results were confirmed in maize (*Zea mays* L.), where Si supplementation increased the expression of salt stress-responsive genes (*ZmSOS1* and *ZmSOS2*) in the root apex and cortex and downregulated the expression of *ZmHKT1*. Authors of this research suggest that these types of transporter systems are involved in enhancing Na^+^ allocation to the leaves via the xylem while decreasing Na^+^ accumulation in the root apex and cortex. Also, Si treatment downregulated the expression of *ZmHKT1* in the root stele, subsequently decreasing Na^+^ unloading from the xylem and increasing the expression of *ZmNHX5*, which is responsible for Na^+^ loading into the leaf vacuole, thereby decreasing the level of Na^+^ in the chloroplasts and improving photosynthetic parameters [76].

### 3.2. Protective Role of Si in Photosynthesis

Plant growth and yield depend heavily on photosynthesis. Salt stress is known to inhibit plant growth and reduce photosynthesis in several ways: ion toxicity, oxidative stress, structural and functional chloroplast damage, osmotic stress-induced reductions in CO_2_ assimilation rate, and inhibition of assimilation product transfer [77,78]. Si has been shown to have a positive impact on shoot growth and net photosynthetic rate in salt-stressed plants, suggesting a potential role in maintaining a high photosynthetic rate under stress conditions [17]. Therefore, several mechanisms by which Si can improve photosynthesis under salt stress were proposed:(1)Reducing ion toxicity and ROS accumulation levels, thus maintaining the structural integrity and function of chloroplast [79,80].(2)Si supplementation increased stomatal conductance, transpiration rate, and stomatal size and number, which resulted in efficient photosynthetic activity under salinity stress [81].(3)Si supplementation can modulate the activities of carbohydrate metabolism enzymes, thus decreasing the content of soluble sugar and starch in leaves but increasing starch content in roots. This alleviated photosynthetic feedback repression in leaves and provided more energy for root growth [82].

In recent years, photosystem II (PSII) was identified as a major salt stress-sensitive component of the photosynthetic system [83]. For instance, in cucumber, salinity stress induced a significant reduction in key chlorophyll fluorescence parameters (such as Fv/Fm (PSII maximum photochemical efficiency), ΦPSII (PSII actual photochemical efficiency), Fv’/Fm’ (PSII effective photochemical efficiency), qP (photochemical quenching coefficient)) and coupled with a substantial increase in NPQ (nonphotochemical quenching coefficient). However, the treatment with Si mitigated these effects, increasing ΦPSII, Fv/Fm, Fv’/Fm’, and qP while decreasing NPQ during salinity stress [84]. Similarly, in maize, Si improved the dry matter yield and water use efficiency (WUE), increased root and shoot K concentration, and decreased shoot Na concentration. Also, Si enhanced the maximum quantum yield of primary photochemistry, thus increasing the efficiency of photosystem II under salinity stress [85]. In salt-stressed mustard plants (*Brassica juncea* (L.) Czern.), Si application decreased Na^+^ uptake and increased the K^+^ and Ca^2+^ concentrations. The content of photosynthetic pigments (Chl a, Chl b, Total Chl, and carotenoids) was also increased after Si treatment. Also, the levels of proline and antioxidants (SOD, ascorbate peroxidase (APX), glutathione (GSH), ascorbate (AsA), and glutathione reductase (GR)) were increased, while the levels of H_2_O_2_, MDA, and electrolyte leakage (EL) rate were decreased [86].

In conclusion, silicon plays a crucial role in enhancing photosynthesis in salt-stressed plants by mitigating Na^+^ ion accumulation, scavenging reactive oxygen species (ROS), and regulating carbohydrate metabolism. The observed improvements in chlorophyll fluorescence parameters highlight Si’s positive impact on the photosynthetic apparatus under stress conditions. However, further in-depth detailed research is imperative to unravel the molecular mechanisms underlying Si’s regulation of ROS and carbohydrate metabolism, including its effects on the expression levels of genes encoding related enzymes. This deeper understanding will contribute to optimising the application of silicon in improving plant resilience to salt stress.

### 3.3. Salt Stress-Mediated Surplus ROS Production and Antioxidant Response

During salinity stress, plants respond by overproducing ROS (hydroxyl radical (HO•), alkoxyl radicals (RO•), singlet oxygen (^1^O_2_), superoxide radical (O_2_^−^•), hydrogen peroxide (H_2_O_2_), and peroxy radicals (ROO•)), which caused oxidative damage to membranes and organelles. The antioxidant systems include enzymatic and nonenzymatic antioxidants. The enzymatic antioxidant system in plants includes SOD, CAT, APX, peroxidase (POD), GR, dehydroascorbate reductase (DHAR), and monodehydroascorbate reductase (MDHAR). The nonenzymatic antioxidants in plants are represented by flavonoids, tocopherols, AsA, carotenoids, and others [87].

Numerous studies have shown that Si could improve ROS scavenging ability by regulating the activities/contents of enzymatic/nonenzymatic antioxidants in plants, and this beneficial effect varies depending on plant species (Figure 3). In sorghum (*Sorghum bicolor* L.), Si application has been proposed to reduce H_2_O_2_ accumulation, which is known as one of the negative regulators of aquaporin, thus alleviating the decrease in hydraulic conductance and increasing water uptake [88]. Moreover, the regulatory pattern varies between plant species and depends on the amount of supplied Si. For example, Si supplementation enhanced the AsA-GSH pathway in two rice cultivars with different salt tolerance, while the ameliorative effect was more pronounced in the sensitive cultivar [89].

The study on *Glycyrrhiza uralensis* (Fisch. ex DC.) showed that the exogenous addition of Si increased SOD and POD activities, reduced MDA concentrations, and increased the K^+^/Na^+^ ratio in stem and leaves compared to salinity stress alone [90]. Another study on *G. uralensis* demonstrated that Si improved plant growth parameters under NaCl and polyethylene glycol (PEG) stresses by modulating stress-response mechanisms. In particular, Si application increased CAT, APX, and GSH content while decreasing membrane permeability and MDA content under combined salt and drought stresses as compared to control plants [91]. Similarly, Si application through rooting media activated antioxidants’ enzymatic (CAT, SOD, POD, APX, and GR) and nonenzymatic (total glutathione, total phenolics, glutathione disulphide (GSSG) and GSH, α-tocopherol, AsA, and proline) defence in barley (*Hordeum vulgare* L.) roots and leaves under salinity stress to mitigate excessive H_2_O_2_ efficiently [92].

In salt-stressed alfalfa (*Medicago sativa* L.), Si supplementation enhanced photosynthetic parameters (photosynthetic rate, transpiration rate, stomatal conductance, and WUE) and total chlorophyll concentration, improved leaf water potential (LWP) in predawn leaf, increased the SOD, CAT, and POD activities, and decreased MDA content. Also, Si limited Na^+^ accumulation while maintaining K^+^ concentration in leaves, thus establishing K^+^/Na^+^ homeostasis to protect the leaves from Na^+^ toxicity and maintain higher chlorophyll retention [93]. These results were also confirmed on the seeds of *Lathyrus odoratus* L. primed with Si and exposed to seawater. Si priming markedly improved seawater-treated seeds’ germination and growth, enhanced water relations, and reduced the loss of photosynthetic pigments and carbohydrates. Also, the Na^+^ accumulation was reduced, while the content and the activities of enzymatic (SOD, APX, and CAT) and nonenzymatic (phenolic and flavonoids) compounds were increased in Si-primed seeds, thus improving oxidative stress (lower MDA and H_2_O_2_ accumulation) parameters and salt stress tolerance [94].

## 4. Role of Osmoprotectants and Polyamines in Si-Mediated Salt Stress-Improving Effects

Recent research has proposed a connection between Si application and the higher levels of polyamine (PA) in leaves and roots during salinity stress (Figure 3). Si application increased the activities of S-adenosylmethionine decarboxylase (SAMDC) and arginine decarboxylase (ADC) while reducing the activity of diamine oxidase (DAO), thus increasing the polyamine levels [95]. Under salinity stress, polyamines also play a role in regulating Na^+^ and K^+^ transport in plants through nonselective ion channels. In sorghum, Si addition increased free and total polyamine levels while decreasing Na^+^ accumulation. Si also balanced polyamine and ethylene metabolism by inhibiting the levels of 1-aminocyclopropane-1-carboxylic acid (ACC), a crucial ethylene precursor, thereby mitigating salt stress [96]. Similar regulatory effects on polyamine levels were observed in salt-stressed cucumbers, indicating Si’s involvement in the reduction in Na^+^ ion toxicity [97].

The comparison of salt stress-sensitive and -tolerant rice cultivars showed that the application of Si and NaCl stress (alone and in combination) increased the PA content and activity of PA synthesising enzymes only in the tolerant cultivar while decreasing both in the sensitive cultivar. NaCl, when applied alone, increased the activities of polyamine degrading enzymes (DAO, polyamine oxidase (PAO) and amino-aldehyde dehydrogenase (AMADH)) in both cultivars, while the effect on the sensitive cultivar was more prominent. However, the application of NaCl in combination with Si decreased the activities of DAO, PAO, and AMADH in the root and shoot. Thus, the results suggested that Si modulates polyamine metabolism (both synthesis and degradation), and these effects are more pronounced in salt stress-tolerant cultivars [98].

Experiments on Chinese liquorice (*Glycyrrhiza uralensis* Fisch. ex DC.) showed that Si promoted carbon and nitrogen metabolism (increased activities of sucrose phosphate synthetase, sucrose synthetase, glutamine synthetase, and nitrate reductase), thus improving plant growth under salt stress [99]. Similarly, various forms of Si nano-biostimulants effectively improved the growth of salt-stressed hemp plants by modulating the amount of soluble sugars (glucose, fructose, sucrose, and galactose) and the expression of stress-responsive genes (*HSP70-2*, *glutathione reductase 2*, and *ethylene response factor 1* (*ERF1*)) [100]. Also, foliar application of Si on salt-stressed basil plants (*Ocimum basilicum* L.) increased proline and soluble sugar content by 352% and 181% compared to salt-stressed control plants without Si treatment [101]. A similar effect was observed also in salt-stressed tomato (*Solanum lycopersicum* L.) plants, where Si, delivered through the foliar and root dipping treatments, increased proline content by 28% [102]. Interestingly, the organ-specific effect of Si on soluble sugar and proline concentrations was demonstrated in salt-stressed rice plants. Thus, Si application increased the content of soluble sugar in the root but decreased it in the shoot. On the contrary, Si treatment decreased proline concentration in the shoot of salt-stressed plants, while the proline content in the root was not affected by Si [103]. These results demonstrated that Si protects plants from salt stress by affecting the metabolism of soluble sugars and proline—crucial osmoprotectants [104,105].

## 5. Role of Phytohormones in Si-Mediated Salt Stress-Ameliorating Effects

Silicon plays a crucial role in enhancing salt tolerance in plants, and recent studies have shed light on the wide transcriptomic changes induced by Si under salinity stress. Limited information is available regarding Si-induced transcriptomic alterations in response to salinity stress, but existing studies provide valuable insights into the gene expression patterns. Among other genes affected by Si under salt stress conditions, many genes associated with the metabolism and signalling of different hormones have been identified. These findings suggested that Si actively regulated levels of abscisic acid (ABA), cytokinins, auxins, gibberellins, and jasmonic and salicylic acids, thus contributing to reduced salt stress-mediated damage and improved salt stress tolerance [10].

Experiments on salt-stressed cucumber demonstrated that Si decreased the content of ABA and indole-3-acetic acid (IAA), alleviated the decrease in cytokinins (trans-zeatin (tZ), trans-zeatin riboside (tZR), isopentenyladenine (iP), and isopentenyladenine riboside (iPR)), and increased salicylic acid (SA) levels, thus normalising it to the control levels. Also, salt stress increased the content of proline (Pro) throughout the experiment, while Si upregulated the salt-induced Pro levels at the beginning of the experiment (days 3–6) but decreased it at the end (days 9–12). Interestingly, the root cytokinin content negatively correlated with Pro content, suggesting their close interaction. Indeed, the application of exogenous Pro under salt stress conditions decreased the expression of *isopentenyl transferase* (*IPT)* genes, while the expression of *cytokinin oxidase* (*CKX)* genes was increased, thus confirming the interplay between Pro and cytokinin metabolism under short-term salt stress [106]. The interplay between Si uptake and cytokinin biosynthesis was further demonstrated in experiments with dark-induced senescence of sorghum and Arabidopsis detached leaves. Therefore, high Si accumulation caused the delay in senescence and was associated with increased expression of cytokinin biosynthesis and signalling genes (*IPT7* and *Arabidopsis response regulator* 5 (*ARR5*)) and higher active cytokinin content (iP). At the same time, the Si-dependent delay in senescence was not observed in *ipt1,3,5,7* leaves [107].

Similarly, Si application improved the growth of salt-stressed seedlings of *G. uralensis* and increased the content of IAA, gibberellic acid (GA_3_), and ABA [108]. However, different results were reported for adult plants (2 years old) in a long-term (150 days) salt stress experiment. Thus, Si application decreased ABA concentration while increasing levels of IAA and GA_3_. In general, the observed effects were beneficial for plant growth and salt stress tolerance but dependent on the NaCl concentration and the plant developmental stage [109]. Si treatment of the date palm (*Phoenix dactylifera* L.) seedling greatly downregulated the concentration of salicylic, jasmonic, and abscisic acids, which were upregulated by salt stress, thus normalising their levels [110].

## 6. Conclusions and Future Prospective

Si supplementation significantly enhances plant salt stress tolerance through multifaceted molecular mechanisms and signalling pathways (Figure 4). The molecular responses to Si supplementation under salt stress involve changes in gene expression, particularly those associated with membrane ion transport, photosynthesis, phytohormone signalling, accumulation and biosynthesis of osmoprotectants, and antioxidant biosynthesis and activity, thus reducing oxidative damage and improving stress resilience. In recent years, Si supplementation emerged as a promising strategy for improving plant salt stress tolerance through its regulatory effects on gene expression, hormonal balance, and physiological responses. The multifaceted nature of Si-mediated effects underscores the need for further research to unravel the intricacies of Si’s molecular mechanisms in different plant species and under varying stress conditions.

Research in the area of plant stress resistance related to Si could benefit from addressing the following questions and exploring specific directions:Mechanistic Understanding:
Transcriptomics and Proteomics: Conduct more extensive and systematic transcriptomic and proteomic studies to unravel the specific genes and proteins influenced by Si supplementation under salt stress conditions. This will provide a more comprehensive understanding of the molecular mechanisms involved.Cellular Signalling Pathways: Investigate the intricate signalling pathways involved in Si-mediated stress tolerance. Explore how Si interacts with established stress signalling molecules, such as calcium ions (Ca^2+^), ROS, and nitric oxide, to modulate stress responses.Hormonal Crosstalk:
Phytohormone Interactions: Elaborate on the crosstalk between Si and different phytohormones. Explore the interconnected regulatory networks, particularly focusing on how Si influences the biosynthesis, signalling, and interaction of known plant hormones.Temporal Dynamics: Investigate the temporal dynamics of hormonal responses to Si supplementation under salt stress. Understand how these responses evolve over time, providing insights into the short-term and long-term effects of Si on salt stress adaptation.Si Uptake and Transport:
Transporter Genes: Explore the regulation of Si transporter genes (*Lsi1* and *Lsi2*) under different stress conditions. Investigate how these transporters contribute to Si uptake and distribution within the plant and their role in salt stress tolerance.Root-Soil Interactions: Study the influence of soil properties and rhizosphere interactions on Si availability and uptake. Determine the optimal conditions for Si supplementation to maximize its effectiveness in improving salt stress resistance.Species-Specific Responses:
Comparative Studies: Conduct comparative studies across a diverse range of plant species to identify commonalities and variations in Si-mediated stress responses. Determine whether the effectiveness of Si supplementation is species-specific and influenced by genetic factors.Crop-Specific Strategies: Tailor Si supplementation strategies to different crops based on their unique physiological and genetic characteristics. Optimise Si application methods, concentrations, and timing for maximum salt stress tolerance in economically important crops.Interaction with Other Abiotic Stresses:
Multiple Stressors: Investigate the combined effects of Si supplementation on plants facing multiple abiotic stresses, such as salinity, drought, and heavy metal toxicity. Understand how Si interacts with these stressors to provide holistic strategies for sustainable agriculture.Environmental Conditions: Explore the influence of varying environmental conditions (temperature, light intensity, humidity, etc.) on the efficacy of Si-mediated salt stress tolerance. Determine the robustness of Si-enhanced resilience under different climate scenarios.Field Trials and Practical Applications:
Field-Scale Studies: Translate laboratory findings into field-scale trials to assess the practical implications of Si supplementation for stress management in real-world agricultural settings. Consider variations in soil types, climates, and cultivation practices. Application of Si depends on the plant growing stage (seed soaking, foliar, root, or soil treatment). Accordingly, working concentration would depend on the Si source (like SiO_2_, Si(OC_2_H_5_)_4_, (K_2_SiO_3_), Na_2_SiO_3_⋅5H_2_O, and so on) and may range from 5 mg L^−1^ to 400 mg L^−1^ [10,111].Economic and Environmental Impact: Evaluate the economic feasibility and environmental impact of large-scale Si supplementation. Consider factors such as cost-effectiveness, potential ecological consequences, and long-term sustainability.


By addressing these research directions, we can deepen our understanding of Si-mediated stress resistance, paving the way for the development of effective and sustainable strategies to enhance plant resilience in the face of diverse environmental challenges.

## Figures and Tables

**Figure 1 plants-13-00525-f001:**
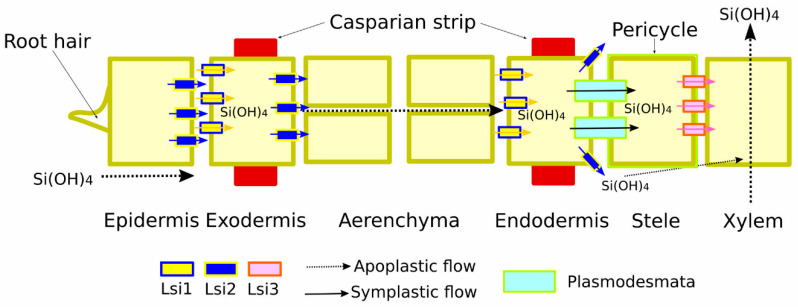
Schematic representation of silicon (Si) transport and xylem loading in rice roots. Si uptake in roots is mediated by two polarly localised transporters—OsLsi1 and OsLsi2. Loading of Si to the xylem is facilitated by OsLsi3, localised to the pericycle. Symplastic flow (cytoplasm-to-cytoplasm) from endodermis to pericycle (via plasmodesmata) and apoplastic flow (along cell walls and extracellular spaces) of Si are shown by solid and dotted arrows; Si transporters—low silicon rice (Lsi) 1-3—depicted with yellow, blue, and pale pink boxes, respectively; plasmodesmata depicted with a pale cyan box.

**Figure 2 plants-13-00525-f002:**
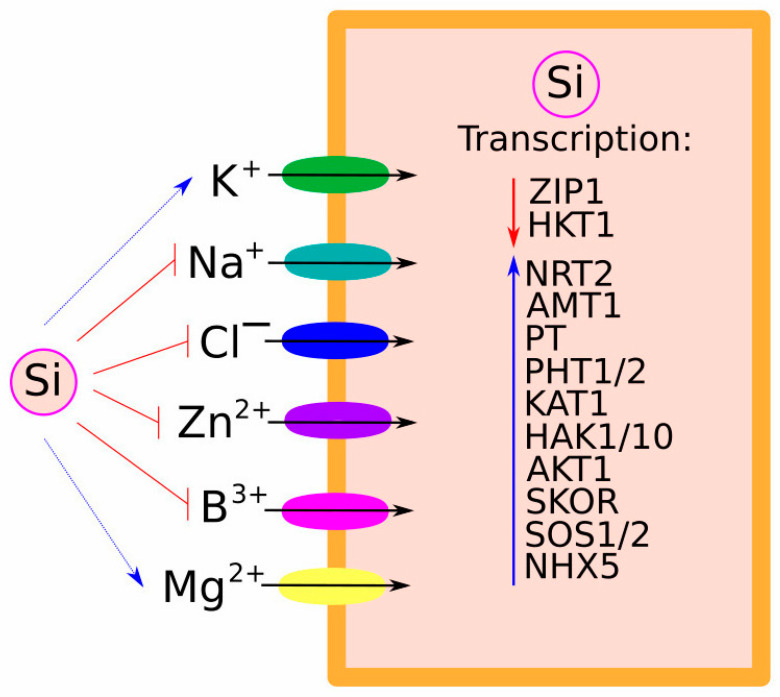
Effect of Si on the expression of other transporters. Under salt stress conditions, Si application reduced the translocation of Na^+^, Cl^−^, Zn^2+^, and B^3+^ (depicted with blunt red arrows) while increasing K^+^ and Mg^2+^ (depicted with dotted blue arrows). Also, Si reduced expression of *ZINC TRANSPORTER 1* (*ZIP1*) and *HIGH-AFFINITY K^+^ TRANSPORTER 1* (*HKT1*) transporters (depicted with red arrow) and increased *phosphate transporter* (*PT*), *high-affinity nitrate transporter protein* (*NRT2*), *high-affinity phosphate transporter 1/2* (*PHT1/1*), *potassium channel protein* (*KAT1*), *potassium transporter protein 1/10* (*HAK1/10*), *high-affinity ammonium transporter protein* (*AMT1*), *K^+^ TRANSPORTER 1, STELAR K^+^ OUTWARD RECTIFIER* (*SKOR*), *SALT OVERLY SENSITIVE 1/2* (*SOS1/2*), and *NA^+^/H^+^ ANTIPORTER 5* (*NHX5*) expression (depicted with blue arrow).

**Figure 3 plants-13-00525-f003:**
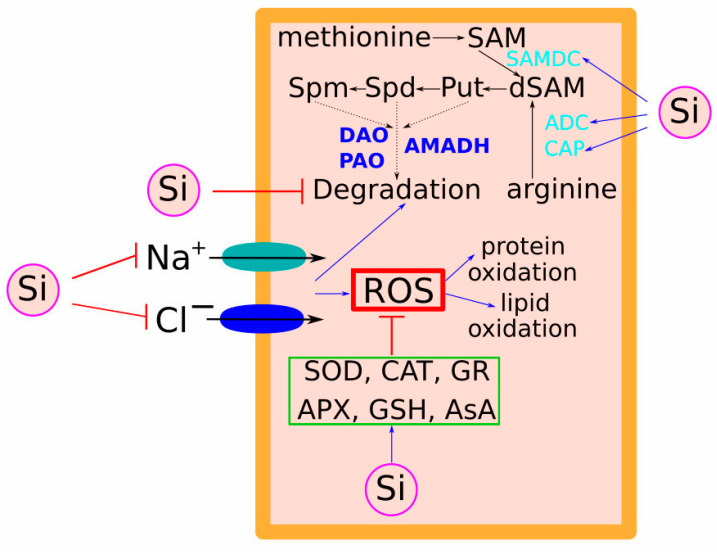
Effect of Si on antioxidant defence and polyamine synthesis. Salt stress increases ROS production, which causes harmful protein and lipid oxidation. Si application reduces salt (NaCl) accumulation and enhances antioxidant production, thus reducing ROS accumulation, protecting against oxidative stress and mitigating ion toxicity. Also, salt stress increased the activities of PA degrading enzymes (diamine oxidase—DAO, polyamine oxidase—PAO, and aminoaldehyde dehydrogenase—AMADH) (depicted in bold blue font), which reduced PA levels and made plants more susceptible to salt stress. Si application, however, decreased activities of PA degrading enzymes and upregulated PA biosynthetic genes (*S-adenosyl-Met-decarboxylase* (*SAMDC*), *arginine decarboxylase* (*ADC*), and *N-carbamoylputrescine amidohydrolase* (*CAP*)) (depicted in cyan font), resulting in high polyamine levels in plant tissues and enhanced salt tolerance. Blue arrows represent positive regulation and blunt red lines are negative.

**Figure 4 plants-13-00525-f004:**
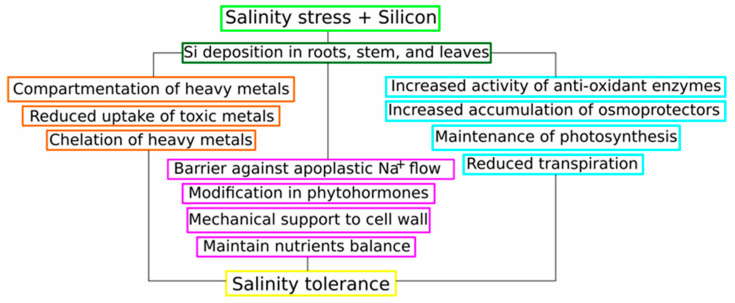
Schematic representation of various beneficial effects of Si supplementation in mitigating salt stress.

## Data Availability

Not applicable.

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
