# Peer review of "The Physiological and Molecular Mechanisms of Silicon Action in Salt Stress Amelioration"

_plants, 2024, doi:10.3390/plants13040525_

Round 1

Reviewer 1 Report

Comments and Suggestions for Authors

The review is well-written and provides a comprehensive summary of the role of Si in processes related to salt stress responses. I have listed a few minor points which could be considered by the authors:

1.         Title: I would recommend adding the word “action” (“The physiological and molecular mechanisms of silicon action in salt stress amelioration”)

2.         It's better to use present simple instead of the past tense used in the Abstract.

3.         Fig. 1: Designate the pericycle cell layer on the scheme.

4.         Line 241: “Thus, Si accumulation in the shoots decreased 67Zn isotope uptake, but no root-to-shoots transport in the wild-type” – please revise the English in the sentence as it is not clear.

5.         Line 242: “Mechanically, it was mediated through”…- I suggest replacing the word “mechanically” with a more suitable one (“indirectly” could be a good alternative).

6.         Figure 2: the expression “Si Transcription:” on the scheme is a bit confusing. I suggest briefly addressing its meaning or removing it.

7.         Lines 254-256: “Si supplementation (in the form of potassium silicate (K2SiO3)) of different poinsettia (Euphorbia pulcherrima Willd. ex Klotzsch) cultivars (a low Si accumulator species) increased the content of Mg and decreased the content of B and Zn in the roots, and only the content of S was increased in the shoots [10.3390/plants8060180].” – “S” (sulfur), I guess the authors have meant “Si”? Please put the correct reference number in the brackets.

8.         Lines: 264-270: “Also, the expression of the main genes responsible for absorption and assimilation of N (nitrate reductase (NR), nitrite reductase (NiR), glutamine synthetase (GS), glutamate synthase (GOGAT), high-affinity nitrate transporter protein (NRT2) and high-affinity ammonium transporter protein (AMT1), P (phosphate transporter (PT), high-affinity phosphate transporter 1 (PHT1), high-affinity phosphate transporter 2 (PHT2) and acid phosphatases (APase)) and K (potassium channel protein (KAT1) and potassium transporter protein (HAK10)) was significantly increased by As, Si and As+Si (Figure 2).” – The mentioned genes in the text do not fully match the ones depicted on the scheme in Fig. 2. So it is not very clear whether Figure 2 is related to the case study in which As and Si were used in combination or it reflects information taken from other studies as well. Therefore, I think that the respective references to each of the listed effects on the transporter genes should be added to the text and also to the figure caption.

9.         Write in full the names of the compounds/proteins at first mention. Such case has been spotted on line 271: “The MDA content was increased by all variants of treatments (As, Si and As+Si), while the activity of SOD, CAT, GPX and GST was stimulated only by As and As+Si, but not by Si”. Some of the abbreviations used in it were written in full later on in the text (line 416).

10.       Line 301: “Further in this section, we discussed the effect of Si application on various systems (metabolism of different ions, hormones, osmoprotectants, antioxidants and photosynthesis) in salt-stressed plants. Further in this section, we discussed the effect of Si supplementation on different physiological processes under salt stress.” – the two sentences repeat the same statement. One of them should be removed or rephrased.

11.       I suggest modifying the heading of Chapter 4 to “Role of osmoprotectants and polyamines in Si-mediated salt-stress-improving effects”. In the paragraph (lines 480-489) which focuses on the role of Si on polyamines metabolism, a reference to Figure 3 should be included. The role of osmoprotectants is discussed only briefly (lines 493-495, related to [95] from the list of references). I advise extending (if possible) this part related to the role of Si supplementation in boosting osmoprotectant accumulation. At least the authors should consider moving the sentence on lines 511-513 to this chapter, as L-Pro is considered to be an osmoprotectant.                        

General comments: It has been observed that an excessive amount of exogenous silica causes nutrient lockout, root damage, and overall low productivity, mainly due to the buildup of salts. Some information on the range in which Si supplementation exerts its beneficial effects in different crops will be a good addition to Chapter 6 “Conclusion and Future Prospective", particularly to “6. Field Trials and Practical Applications”. It may be helpful from a practical point of view. Including brief information on the safety of Si supplementation in agriculture regarding its possible implications on the food chain will be also useful.

Author Response

Dear Editor and Reviewers,
We greatly appreciate your critical evaluation of our manuscript and helpful comments. Our reply to your comments would be provided point by point, where “A” stands for “Authors”, and “L” for “Lines”, where changes have been implemented.

____________________________________________________________________________

The review is well-written and provides a comprehensive summary of the role of Si in processes related to salt stress responses. I have listed a few minor points which could be considered by the authors:

  1. Title: I would recommend adding the word “action” (“The physiological and molecular mechanisms of silicon action in salt stress amelioration”)

A: The title was modified as suggested.

  1.        It's better to use present simple instead of the past tense used in the Abstract.

A: The abstract was modified as suggested.

  1. Fig. 1: Designate the pericycle cell layer on the scheme.

A: Figure 1 was modified as suggested.

  1. Line 241: “Thus, Si accumulation in the shoots decreased 67Zn isotope uptake, but no root-to-shoots transport in the wild-type” – please revise the English in the sentence as it is not clear.

A: Thank you for the valuable comment. Indeed, the crucial part of the sentence was missing. The sentence was modified, please, see L251.

  1. Line 242: “Mechanically, it was mediated through”…- I suggest replacing the word “mechanically” with a more suitable one (“indirectly” could be a good alternative).

A: The sentence was modified as suggested. Please, see L252.

  1. Figure 2: the expression “Si Transcription:” on the scheme is a bit confusing. I suggest briefly addressing its meaning or removing it.

A: The title of figure 2 was modified. Please, see L255.

  1. Lines 254-256: “Si supplementation (in the form of potassium silicate (K2SiO3)) of different poinsettia (Euphorbia pulcherrima Willd. ex Klotzsch) cultivars (a low Si accumulator species) increased the content of Mg and decreased the content of B and Zn in the roots, and only the content of S was increased in the shoots [10.3390/plants8060180].” – “S” (sulfur), I guess the authors have meant “Si”? Please put the correct reference number in the brackets.

A: The sentence was clarified (“S” stands for sulphur). The reference was modified according to the Plants reference style.

  1. Lines: 264-270: “Also, the expression of the main genes responsible for absorption and assimilation of N (nitrate reductase (NR), nitrite reductase (NiR), glutamine synthetase (GS), glutamate synthase (GOGAT), high-affinity nitrate transporter protein (NRT2) and high-affinity ammonium transporter protein (AMT1), P (phosphate transporter (PT), high-affinity phosphate transporter 1 (PHT1), high-affinity phosphate transporter 2 (PHT2) and acid phosphatases (APase)) and K (potassium channel protein (KAT1) and potassium transporter protein (HAK10)) was significantly increased by As, Si and As+Si (Figure 2).” – The mentioned genes in the text do not fully match the ones depicted on the scheme in Fig. 2. So it is not very clear whether Figure 2 is related to the case study in which As and Si were used in combination or it reflects information taken from other studies as well. Therefore, I think that the respective references to each of the listed effects on the transporter genes should be added to the text and also to the figure caption.

A:  There are several studies referred to the figure 2 (please see also previous paragraph of section 2.2, and section 3.1).

All the mentioned genes (L264-270) were investigated in the cited paper [62]. Also, all 3 variants (As alone, Si alone and combination As+Si) up-regulated the mentioned genes. This point was further clarified in the text, please, see L283.

  1. Write in full the names of the compounds/proteins at first mention. Such case has been spotted on line 271: “The MDA content was increased by all variants of treatments (As, Si and As+Si), while the activity of SOD, CAT, GPX and GST was stimulated only by As and As+Si, but not by Si”. Some of the abbreviations used in it were written in full later on in the text (line 416).

A: Used abbreviation were decrypted, please, see L285-286.

  1. Line 301: “Further in this section, we discussed the effect of Si application on various systems (metabolism of different ions, hormones, osmoprotectants, antioxidants and photosynthesis) in salt-stressed plants. Further in this section, we discussed the effect of Si supplementation on different physiological processes under salt stress.” – the two sentences repeat the same statement. One of them should be removed or rephrased.

A: The second sentence was removed. Please, see L320-321.

  1. I suggest modifying the heading of Chapter 4 to “Role of osmoprotectants and polyamines in Si-mediated salt-stress-improving effects”.

A: The title of the section 4 was modified as suggested. Please, see L491

In the paragraph (lines 480-489) which focuses on the role of Si on polyamines metabolism, a reference to Figure 3 should be included.

A: Reference to Figure 3 was added.

The role of osmoprotectants is discussed only briefly (lines 493-495, related to [95] from the list of references). I advise extending (if possible) this part related to the role of Si supplementation in boosting osmoprotectant accumulation. At least the authors should consider moving the sentence on lines 511-513 to this chapter, as L-Pro is considered to be an osmoprotectant.

A: The section 4 was further expanded. Please see L522-531.

While the suggested sentence (511-513) is more related to hormonal regulation, we would prefer to keep in the section 5.

General comments: It has been observed that an excessive amount of exogenous silica causes nutrient lockout, root damage, and overall low productivity, mainly due to the buildup of salts. Some information on the range in which Si supplementation exerts its beneficial effects in different crops will be a good addition to Chapter 6 “Conclusion and Future Prospective", particularly to “6. Field Trials and Practical Applications”. It may be helpful from a practical point of view. Including brief information on the safety of Si supplementation in agriculture regarding its possible implications on the food chain will be also useful.

A: We thank the reviewer for a valuable comment. However, to the best of our knowledge, there are no articles reporting negative results. On one side, this fact may be related to the general trend in modern science to report only positive/beneficial results. Alternatively, authors may be just carefully selecting Si concentration/ Si source and delivery system. Also, none of the cited papers have analysed any long-term related effects of Si supplementation on the environment, elements of the food chain or pathogens.

Used Si parameters have been analysed in several recent reviews (for example, here [10] and [111]). These data were briefly added to the suggested section 6.

Reviewer 2 Report

Comments and Suggestions for Authors

This is an interesting review article which is close to being fully comprehensive.

The text is well written and there are no substantial issues with the English.

The main problem I identified is the extensive use of abbreviations which are not always spelled out at first use thereby rendering understanding of some statements rather laborious. In the absence of an abbreviation list (which would have anyway been of little use given its length) the authors must check and verify that all abbreviations are explained at first use.

Another point needing amendment is the colours used in Figure 2 and 3. For instance, Fig. 2 could be clearer if different arrow colours (or styles) were used to distinguish between elements whose translocation is reduced vs. increased.

In both Figs. 2 and 3 it would be better to prefer red/blue instead of blue/green, as advocated internationally to avoid problems for Daltonian readers.

The text in lines 91-109 could be clearer. The mechanisms underlying Si uptake and transport is certainly complex and it is not with these many unexplained abbreviations that readers will better understand it.

§ 113-128 = check that all gene names are italicised

With the near-complete literature survey undertaken by the authors, I found it a pity that no mention (or nearly) was made to the use of Si nanoparticles in plants within the context of salt and osmotic stress. This is also an area of intense interest lately and including some information about it would render this review paper fully comprehensive.

In this respect, I am including below a non-exhaustive list of potential references that might be used to write up a short section on this:

Avestan S, Naseri L, Barker AV (2017) Evaluation of nanosilicon dioxide and chitosan on tissue culture of apple under agar-induced osmotic stress. J Plant Nutr 40:2797–2807. https://doi.org/10.1080/01904167.2017

Behboudi F, Tahmasebi Sarvestani Z, Kassaee MZ, Modares Sanavi S, Sorooshzadeh A (2018) Improving growth and yield of wheat under drought stress via application of SiO2 nanoparticles. J Agric Sci Technol 20:1479–1492

Sabaghnia N, Janmohammadi M (2015) Effect of nano-silicon particles application on salinity tolerance in early growth of some lentil genotypes. Ann Univ Mariae Curie-Skiodowska C Biol 69:39–55. https://doi.org/10.1515/umcsbio-2015-0004

Siddiqui MH, Al-Whaibi MH, Faisal M, Al Sahli AA (2014) Nano‐silicon dioxide mitigates the adverse effects of salt stress on Cucurbita pepo L. Environ toxicol chem 33:2429–2437. https://doi.org/10.1002/etc.2697

Yassen A, Abdallah E, Gaballah M, Zaghloul S (2017) Role of silicon dioxide nano fertilizer in mitigating salt stress on growth, yield and chemical composition of cucumber (Cucumis sativus L). Int J Agric Res 12:130–135. https://doi.org/10.3923/ijar.2017.130.135

 I could also suggest some additional general references about the use of Si in agriculture, as follows:

 Artyszak A (2018) Effect of silicon fertilization on crop yield quantity and quality—A literature review in Europe. Plants 7:54. https://doi.org/10.3390/plants7030054

Debona D, Rodrigues FA, Datnoff LE (2017) Silicon’s role in abiotic and biotic plant stresses. Annu Rev Phytopathol 55:85–107. https://doi.org/10.1146/annurev-phyto-080516-035312

Gong Hj C, Km C, Sm GW, Zhang Cl (2003) Effects of silicon on growth of wheat under drought. J Plant Nutr 26:1055–1063. https://doi.org/10.1081/PLN-120020075

Ma JF (2004) Role of silicon in enhancing the resistance of plants to biotic and abiotic stresses. Soil Sci Plant Nutr 50:11–18. https://doi.org/10.1080/00380768.2004.10408447

Ranjan A, Sinha R, Bala M, Pareek A, Singla-Pareek SL, Singh AK (2021) Silicon-mediated Abiotic and biotic stress mitigation in plants: underlying mechanisms and potential for stress resilient agriculture. Plant Physiol Biochem 163:15–25. https://doi.org/10.1016/j.plaphy.2021.03.044

Yan GC, Nikolic M, Ye MJ, Xiao ZX, Liang YC (2018) Silicon acquisition and accumulation in plant and its significance for agriculture. J Integr Agric 17:10: 2138–2150. https://doi.org/10.1016/S2095-3119(18)62037-4

This being said, I believe this manuscript could be accepted pending revision.

Author Response

Dear Editor and Reviewers,
We greatly appreciate your critical evaluation of our manuscript and helpful comments. Our reply to your comments would be provided point by point, where “A” stands for “Authors”, and “L” for “Lines”, where changes have been implemented.

____________________________________________________________________________

This is an interesting review article which is close to being fully comprehensive.

The text is well written and there are no substantial issues with the English.

The main problem I identified is the extensive use of abbreviations which are not always spelled out at first use thereby rendering understanding of some statements rather laborious. In the absence of an abbreviation list (which would have anyway been of little use given its length) the authors must check and verify that all abbreviations are explained at first use.

A: used abbreviations were checked and decrypted at the first use.

Another point needing amendment is the colours used in Figure 2 and 3. For instance, Fig. 2 could be clearer if different arrow colours (or styles) were used to distinguish between elements whose translocation is reduced vs. increased.

In both Figs. 2 and 3 it would be better to prefer red/blue instead of blue/green, as advocated internationally to avoid problems for Daltonian readers.

A: Figures 2 and 3 were modified as suggested.

The text in lines 91-109 could be clearer. The mechanisms underlying Si uptake and transport is certainly complex and it is not with these many unexplained abbreviations that readers will better understand it.

A: There are no undefined abbreviations on the L91-109. It is clearly indicated in the text that  WIGR, WVAR and AIGR and so are amino acids. Used code (1 and 3 letters) is universal and internationally accepted way to refer to a particular amino acid. The designation for water molecules in the structure (L107) was further clarified.

  • 113-128 = check that all gene names are italicised

A: Checked. All gene names are italicised.

With the near-complete literature survey undertaken by the authors, I found it a pity that no mention (or nearly) was made to the use of Si nanoparticles in plants within the context of salt and osmotic stress. This is also an area of intense interest lately and including some information about it would render this review paper fully comprehensive.

In this respect, I am including below a non-exhaustive list of potential references that might be used to write up a short section on this:

Avestan S, Naseri L, Barker AV (2017) Evaluation of nanosilicon dioxide and chitosan on tissue culture of apple under agar-induced osmotic stress. J Plant Nutr 40:2797–2807. https://doi.org/10.1080/01904167.2017

Behboudi F, Tahmasebi Sarvestani Z, Kassaee MZ, Modares Sanavi S, Sorooshzadeh A (2018) Improving growth and yield of wheat under drought stress via application of SiO2 nanoparticles. J Agric Sci Technol 20:1479–1492

Sabaghnia N, Janmohammadi M (2015) Effect of nano-silicon particles application on salinity tolerance in early growth of some lentil genotypes. Ann Univ Mariae Curie-Skiodowska C Biol 69:39–55. https://doi.org/10.1515/umcsbio-2015-0004

Siddiqui MH, Al-Whaibi MH, Faisal M, Al Sahli AA (2014) Nano‐silicon dioxide mitigates the adverse effects of salt stress on Cucurbita pepo L. Environ toxicol chem 33:2429–2437. https://doi.org/10.1002/etc.2697

Yassen A, Abdallah E, Gaballah M, Zaghloul S (2017) Role of silicon dioxide nano fertilizer in mitigating salt stress on growth, yield and chemical composition of cucumber (Cucumis sativus L). Int J Agric Res 12:130–135. https://doi.org/10.3923/ijar.2017.130.135

 I could also suggest some additional general references about the use of Si in agriculture, as follows:

 Artyszak A (2018) Effect of silicon fertilization on crop yield quantity and quality—A literature review in Europe. Plants 7:54. https://doi.org/10.3390/plants7030054

Debona D, Rodrigues FA, Datnoff LE (2017) Silicon’s role in abiotic and biotic plant stresses. Annu Rev Phytopathol 55:85–107. https://doi.org/10.1146/annurev-phyto-080516-035312

Gong Hj C, Km C, Sm GW, Zhang Cl (2003) Effects of silicon on growth of wheat under drought. J Plant Nutr 26:1055–1063. https://doi.org/10.1081/PLN-120020075

Ma JF (2004) Role of silicon in enhancing the resistance of plants to biotic and abiotic stresses. Soil Sci Plant Nutr 50:11–18. https://doi.org/10.1080/00380768.2004.10408447

Ranjan A, Sinha R, Bala M, Pareek A, Singla-Pareek SL, Singh AK (2021) Silicon-mediated Abiotic and biotic stress mitigation in plants: underlying mechanisms and potential for stress resilient agriculture. Plant Physiol Biochem 163:15–25. https://doi.org/10.1016/j.plaphy.2021.03.044

Yan GC, Nikolic M, Ye MJ, Xiao ZX, Liang YC (2018) Silicon acquisition and accumulation in plant and its significance for agriculture. J Integr Agric 17:10: 2138–2150. https://doi.org/10.1016/S2095-3119(18)62037-4

This being said, I believe this manuscript could be accepted pending revision.

A: We thank the reviewer for this suggestion. However, full coverage of suggested topics (osmotic and drought stress) would require the writing of another paper. Also, this review is focused on the recent data, so the old papers (dated, for example, by 2003 and 2004) would not be included. Suggested papers relevant to our topic (salt stress) have been added to the reference list, please see [7, 8, 9, 11].

Reviewer 3 Report

Comments and Suggestions for Authors

plants-2868495

Title: The physiological and molecular mechanisms of silicon in salt stress amelioration

Authors: Siarhei A. Dabravolski and Stanislav V. Isayenkov

This review is well organized and comprehensively described and well written and also has some innovations. Additionally, I believe that the thematic content of this journal is also suitable for Plants' international audience.

I encourage the editor to accept the publication of this review in its current state.

Author Response

Dear Editor and Reviewers,
We greatly appreciate your critical evaluation of our manuscript and helpful comments. Our reply to your comments would be provided point by point, where “A” stands for “Authors”, and “L” for “Lines”, where changes have been implemented.

____________________________________________________________________________

This review is well organized and comprehensively described and well written and also has some innovations. Additionally, I believe that the thematic content of this journal is also suitable for Plants' international audience.

I encourage the editor to accept the publication of this review in its current state.

A: we wish to thank the reviewer for the high evaluation of our work
